# Poor Outcomes of Girdlestone Resection Arthroplasty in Injection Drug Users: A Retrospective Study

**DOI:** 10.3390/antibiotics13080782

**Published:** 2024-08-21

**Authors:** Henry T. Shu, Diane Ghanem, Oscar Covarrubias, Zaid Elsabbagh, Alice J. Hughes, Rachel B. Sotsky, Janet D. Conway, Jamie Ferguson, Greg M. Osgood, Babar Shafiq

**Affiliations:** 1Department of Orthopaedic Surgery, Johns Hopkins Hospital, Baltimore, MD 21287, USA; dghanem1@jh.edu (D.G.); ocovarrubias@uoi.com (O.C.); zelsabb1@jh.edu (Z.E.); gosgood2@jhmi.edu (G.M.O.); bshafiq2@jhmi.edu (B.S.); 2Department of Orthopaedic Surgery, Johns Hopkins Bayview Medical Center, Baltimore, MD 21224, USA; ahughe34@jhmi.edu (A.J.H.); rsotsky1@jhmi.edu (R.B.S.); 3International Center for Limb Lengthening, Rubin Institute for Advanced Orthopaedics, Sinai Hospital, Baltimore, MD 21215, USA; jconway@lifebridgehealth.org; 4Bone Infection Unit, Nuffield Orthopaedic Centre, Oxford University Hospitals, Oxford OX3 7LD, UK; jamieferguson@doctors.org.uk

**Keywords:** Girdlestone resection arthroplasty, injection drug use, septic hip arthritis, antibiotic spacer, clinical outcomes

## Abstract

This retrospective cohort study aims to investigate the clinical outcomes of Girdlestone resection arthroplasty (GRA) in injection drug users with septic hip arthritis. Patients who underwent primary GRA for septic hip arthritis secondary to injection drug use at two academic trauma centers from 2015 to 2023 were retrospectively reviewed. Patient demographics, surgical details, and follow-up outcomes, including patient-reported outcome measures, were collected and analyzed. The cohort included 15 patients, with a mean age of 44 ± 11 years and a mean follow-up period of 25 ± 20 months. Among the 15 patients, overall mortality was 27%, and only 4 patients underwent total hip arthroplasty (THA) following GRA. Infection resolution rates were significantly higher in patients who received an antibiotic spacer (75% vs. 0%, *p* = 0.048). GRA in injection drug users is associated with high mortality and low conversion rates to THA. The use of an antibiotic spacer during GRA significantly improves infection resolution rates. Larger studies are required to determine the optimal management strategies for this patient population.

## 1. Introduction

Girdlestone resection arthroplasty (GRA) is an eponymous term for resection of the femoral head and neck. While initially described by Gathorne Robert Girdlestone in the early 20th century for tuberculosis-infected hips, GRA is now primarily performed for elderly patients with infected total hip arthroplasty (THA) who are unable to receive a revision THA [1,2,3,4,5]. Nevertheless, GRA may still be performed in younger patients who present with osteomyelitis of the native hip.

One such population that is at risk for hip osteomyelitis is patients with injection drug use, where the hematogenous seeding of bacteria may lead to osteomyelitis of the hip [6,7]. These patients often require GRA due to severe bony lysis, high re-infection risk, and high rates of drug relapse, making THA impractical [6,7,8,9,10,11]. Moreover, as these patients are generally younger than the typical patients undergoing THA, it is important to understand the long-term implications of performing GRA in these patients.

With rising injection drug use rates globally, orthopedic surgeons are more likely to encounter patients with septic hips in this demographic, necessitating a thorough understanding of GRA outcomes [12,13,14,15]. Despite the long history of the GRA, the quality of life and functional outcomes post-GRA in younger patients have been poorly characterized [3,7,16,17,18,19]. Existing studies predominantly address secondary procedures for prosthetic joint infections, with limited data on primary GRA outcomes in this specific population [3,7,16,17,18,19].

Therefore, this study aims to describe the outcomes of primary GRA for hip osteomyelitis in injection drug users and compare outcomes between those who received an antibiotic spacer and those who did not. We hypothesize that antibiotic spacer use correlates with better patient-reported outcome measures (PROMs) and that overall, there will be a low conversion rate to THA with a high revision rate for those who did receive THAs.

## 2. Results

### 2.1. Patient Demographics and Characteristics

Out of 51 patients who received a GRA (CPT-27122) at the authors’ two institutions, 15 (29%) received a primary GRA for infection secondary to injection drug use (Figure 1). Inclusion was based on the documentation of injection drug use in patient’s history and radiographic, CT imaging, and/or MRI signs of infection, elevated inflammatory markers [C-reactive protein (CRP), erythrocyte sedimentation rate (ESR), complete blood count (CBC)], presence of purulence or infection in operative reports, and/or positive intraoperative cultures. Of the 15 patients, 10 (67%) did not receive an antibiotic spacer at the time of GRA. The mean age was 44 ± 11 years, and BMI was 23.1 ± 5.3. The mean follow-up was 25 ± 20 months (range 1–69 months) (32 ± 20 months, range 6–69, when excluding deceased patients). Patients who underwent GRA without an antibiotic spacer had significantly higher rates of previously diagnosed conditions compared to those who received a spacer (60% vs. 0%, *p* = 0.044 Table 1). No patients had hepatitis B infection. Nine (60%) patients had untreated, or chronic, HCV infections, and three (20%) patients had a previous history of endocarditis. Patients who did not receive an antibiotic spacer had a significantly greater percentage of previously diagnosed psychiatric conditions (60% versus 0%, *p* = 0.044). Notably, all patients had active tobacco use at the time of their primary GRA.

### 2.2. Sepsis, Blood Cultures, and Mortality

Among the 15 patients, 4 (27%) patients met SIRS criteria per Sepsis-3 consensus definitions [20]. Positive blood cultures were found in four (27%) patients, primarily growing methicillin-sensitive *Staphylococcus aureus* (MSSA) and group A *Streptococcus* species (GAS), all of which were consistent with subsequent intraoperative cultures. Of these four patients with positive blood cultures, only one met the SIRS criteria on presentation. No septic patients or patients with positive blood cultures had seeding of other joints during their admission for GRA.

Overall mortality was 27% (4/15), with a median time to death of 78 (60–227) days. Three patients who did not receive antibiotic spacers died, whereas one patient who received an antibiotic spacer died (Table 1, *p* = 0.180). Two (13%) patients were paraplegic, both of whom did not receive an antibiotic spacer. Deidentified descriptions of each patient are available in Appendix A. Regarding the deaths, one patient died from a subsequent opioid overdose, one patient died from massive PE 3 months after GRA, and two patients had unknown causes of death.

### 2.3. Microbiology—Hip Aspirations

Nine (60%) patients received hip aspirations prior to their GRA, with three of the hip aspirates being negative for growth. Of these nine hip aspirates, two grew methicillin-resistant *Staphylococcus aureus* (MRSA), two grew MSSA, one grew GAS, and one grew both *Micrococcus luteus* and *Cutibacterium acnes*. These hip aspirates were generally consistent with intraoperative cultures, except for the sample that grew *Micrococcus luteus* and *Cutibacterium acnes*, in which the final intraoperative culture was negative for growth. This may have been due to contamination or due to starting antibiotic and antifungal therapy pre-aspiration.

### 2.4. Microbiology—Intraoperative Cultures

All patients had intraoperative deep tissue sent for culture. Of the intraoperative cultures, three samples grew MRSA, four grew MSSA, and two cases had polymicrobial infection. Of the two cases with polymicrobial infection, the first grew a Vancomycin-resistant Enterococcus species (VRE), *Candida albicans*, and *Pantoea septica*. The second grew GAS, *Enterococcus faecalis*, *Enterococcus faecium*, *Proteus mirabilis*, *Arcanobacterium hemolyticum*, and *Brevibacterium* species. The full descriptions of antibiotic therapy for each patient are described in Appendix A. All antibiotic therapy was dictated based on the intraoperative cultures. In the case of a negative intraoperative culture, antibiotic therapy was tailored based on hip aspirate cultures (if performed) or blood cultures. If all cultures were negative, broad spectrum empirical antibiotic therapy was continued, typically with coverage of MRSA and other pathogenic skin flora (Appendix A).

### 2.5. Objective Outcomes

Following GRA, four patients received a THA, of which three (75%) required revisions. Among the nine patients analyzed for outcomes, seven (78%) were ambulatory and six (67%) had complete resolution of their hip infection (Table 2). Of the three revision THAs, two were due to septic loosening or infected THA in the setting of continued injection drug use. These two infected THAs were treated with an explant and placement of a subsequent antibiotic spacer. Unfortunately, one of those two patients fractured their femur below the antibiotic spacer and had subsequent open reduction and internal fixation, nonunion, and ultimately resection of the proximal femur with removal of all implants. The one revision not due to infection was caused by recurrent dislocations and was successfully treated with a revision of the acetabular cup to a larger size.

When comparing those who received antibiotic spacers to those who did not, the rates of infection resolution after a single admission were significantly greater in the antibiotic spacer group (0% vs. 75%, *p* = 0.048). Additionally, patients who did not receive an antibiotic spacer had a significantly greater rate of readmission for continued infection in the ipsilateral hip (100% vs. 25%, *p* = 0.048). No other objective outcome measures were significantly different between those who received antibiotic spacers and those who did not (Table 2).

### 2.6. Patient-Reported Outcome Measures

Of the nine eligible patients, only four (44%) patients completed PROMs (Table 3). Given the low response rates, no statistical analysis was performed. The EQ-5D-5L mobility dimension was associated with the greatest difficulty (mean 3.3 ± 1.3).

### 2.7. In-Hospital Psychiatric Interventions

During hospitalization, 80% (12/15) of patients were consulted by the substance use disorder (SUD) team, of which 67% (8/12) agreed to inpatient SUD rehabilitation. Psychiatric consults were significantly more common in the non-antibiotic spacer group (53% vs. 0%, *p* = 0.007, Table 4).

## 3. Discussion

The primary purpose of this study is to describe the outcomes following primary GRA for septic arthritis of the hip in injection drug users. This study found a high postoperative mortality rate (27%) and low rate of conversion to THA (44%), with a large proportion of THA conversions requiring revision due to infection. Nevertheless, 67% of patients had complete resolution of their hip infections. While the use of an antibiotic hip spacer was associated with a higher rate of infection resolution (75% vs. 0%, *p* = 0.048, Table 2), this may reflect the selection of patients with less severe disease rather than the effect of the spacer itself.

There is very limited literature on outcomes in injection drug users who receive primary GRA, with Maguire et al. reporting a case without long-term data [19]. Other recent studies regarding primary GRA in the setting of infection and/or fracture have shown similar rates of mortality and limited functional outcomes as compared to this study [17,18]. The poor outcomes in the injection drug use population are likely due to high rates of substance use relapse over the course of an addiction [11,21] and high rates of psychiatric and medical comorbidities in this population despite their generally young age [2,22,23].

The predominance of skin flora, such as MSSA, MRSA, and GAS, is consistent with previous studies, underscoring the need for targeted antibiotic therapy [6,8]. Notably, polymicrobial infections in paraplegic patients suggest a need for vigilance regarding opportunistic infections from the gastrointestinal tract [24]. A higher rate of opportunistic infection and unusual organisms should be considered in this patient group due to the recurrent bacteremia associated with contaminated needle use, as well as the associated risk of immunocompromise related to poor general health, malnutrition, and the association with HIV and hepatitis C.

Furthermore, some unique considerations in managing infection in injection drug users include poor venous access and the potential for indwelling lines to be used for continued substance abuse. Thus, physicians who manage these patients should seek to convert to oral antibiotics whenever possible, as previous trials have suggested that oral antibiotics are noninferior to IV antibiotics for complex bone infections [25]. Notably, every patient in this study received local antibiotic therapy, which was typically vancomycin and tobramycin, with the dose determined at the discretion of the surgeon (Appendix A). This has the advantage of allowing a high local level of antibiotics at the infection and circumvents some of the issues relating to concordance with ongoing antibiotic therapy that may be an issue in this group.

As the rates of injection drugs continue to dramatically increase globally, there is a need to better understand the outcomes following treatment of infection secondary to injection drug use [14,15]. Given the high relapse rates and complex infection profiles, multidisciplinary management involving infectious disease specialists, psychiatrists, and substance use disorder experts is crucial for improving outcomes. Nevertheless, this study highlights the need for improved management strategies for injection drug users with septic hip arthritis, emphasizing the importance of comprehensive, multidisciplinary care.

## 4. Limitations

The small sample size of this study is a severe limitation, impacting the generalizability and statistical power of the study’s findings. Furthermore, the retrospective nature of this study introduces selection bias and limits the ability to establish causality regarding the efficacy of antibiotic spacers in facilitating single-stage GRA. Selection bias is evident in patients who did not receive antibiotic spacers having greater rates of psychiatric diagnoses, skewing outcomes. Furthermore, in cases of severe osteomyelitis with extensive bony loss, the placement of antibiotic spacers may not be feasible due to the insufficient stability of the acetabulum and proximal femur post-debridement. Additionally, collecting PROMs in this patient population is challenging due to housing instability and unreliable contact information, resulting in only four patients completing PROMs. This severely limits the generalizability of the PROMs in this study. In-office follow-ups are also difficult in this patient population, further reducing the likelihood of THA conversion because of patient noncompliance.

## 5. Methods

Following Institutional Review Board approval (JHM IRB00391703), a retrospective chart review was performed at two United States academic trauma centers between January 2015 and December 2023. This investigation was conducted in accordance with the Declaration of Helsinki of 1975. A STROBE checklist was submitted with this study. Patients who underwent GRA (CPT-27122) were identified through electronic medical records. Inclusion criteria were primary GRA for infected hip secondary to injection drug use. Patients with secondary GRA or pre-existing THA were excluded. No sample size estimation was performed.

### 5.1. Surgical Technique

All patients received a posterior approach to the hip, with a lateral window approach if the infection extended into the pelvis. The decision to use an antibiotic cement spacer was made by the attending surgeon based on the bone integrity of the acetabulum and proximal femur after debridement. Acetabular reaming was only performed if a spacer was placed; otherwise, cartilage was fully removed via curettage. Either a Depuy Synthes (Raynham, MA, USA) Prostalac Hip System or a Zimmer Biomet (Warsaw, IN, USA) Taperloc or Echo stem with antibiotic cement was used. A total of 40 g of Palacos (Heraeus Medical, Hanaus, Germany) cement with 3 g of vancomycin and 1.2 g of tobramycin was used in all cases for cement fixation. Figure 2 and Figure 3 illustrate severe bony involvement that precluded spacer placement, whereas Figure 4 shows an antibiotic spacer. The use of local antibiotic therapy was also at the discretion of the surgeon at the time of GRA. All surgeons took intraoperative deep cultures with fresh instruments at the time of GRA. A minimum of three samples were obtained, allowing for aerobic, anaerobic, and fungal cultures.

### 5.2. Primary Outcomes

The primary outcomes included mortality, conversion to THA, revision THA, ability to ambulate at final follow-up, and complete resolution of infection. A revision THA was defined as any procedure requiring a return to the operating room for a complication of the THA, including polyethylene exchange. Complete resolution of infection was defined as the eradication of infection with no further radiographic, laboratory, or clinical evidence of continued infection in the absence of any suppressive antibiotics. Laboratory studies routinely ordered for infection at both institutions included CRP, ESR, and CBC.

### 5.3. Secondary Outcomes

Secondary outcomes included patient demographics, social history, and medical comorbidities. Demographics collected included age, body mass index (BMI), and gender. Social history included active injection drug use at initial hospitalization (defined as injection drug use within one week of hospitalization), tobacco use, and stable housing (defined as having a permanent place of residence). Medical comorbidities included chronic hepatitis C virus (HCV) infection, history of endocarditis, paraplegia, other diagnosed psychiatric disorders (defined as any formally diagnosed psychiatric disorder in the medical chart other than SUD), and the presence of systemic inflammatory response syndrome (SIRS), as per the Sepsis-3 consensus definitions [20].

### 5.4. Collection of Patient-Reported Outcome Measures

PROMs were collected via phone-administered surveys with verbal consent. Measures included the EuroQoL 5-dimension 5-level (EQ-5D-5L) [26,27], the Numerical Rating Scale (NRS) for pain [28], and the Oxford Hip Score [29], with deceased or paraplegic individuals being excluded. The visual analog scale (VAS) from the EQ-5D-5L was omitted as the surveys were completed verbally via phone.

### 5.5. Psychiatric Interventions

The rates of SUD team consults, psychiatry consults, and peer recovery consults were also recorded. At the institutions where this study was performed, the SUD teams included a pharmacist and a physician who specialized in prescribing methadone and/or buprenorphine for opioid use disorder and treatment of withdrawal syndromes. Additionally, the peer recovery consult service connects patients with community outreach activists, many of whom were previous substance users who have successfully experienced recovery and provide counseling and support for patients with active SUD. The rates of patients agreeing to inpatient SUD treatment were also recorded.

### 5.6. Statistical Analysis

Patients were included for outcomes analysis if they had a minimum of 1-year follow-up. Patients who were paraplegic were excluded from the analysis regarding THA, ambulation, and PROMs. Final follow-up was considered as the date of the last in-person orthopedic exam or the date the PROM survey was completed. Continuous variables were assessed for normality via the Shapiro–Wilk test. Continuous variables were compared with the Mann–Whitney U-test for non-normal distribution or *t*-tests for normal distribution. Continuous variables were reported as mean ± standard deviation (SD) or median [interquartile range (IQR)] as appropriate. Categorical variables were compared with Fisher’s exact test. An α value of 0.05 was used. *p*-value multiplicity correction was not performed for two reasons: (1) the small sample size of this study results in a far greater risk for type 2 error as opposed to type 1 error, even with multiple testing. (2) This retrospective cohort study was inherently exploratory, and its findings are meant to spur larger prospective studies.

## 6. Conclusions

GRA in patients with septic hip arthritis secondary to injection drug was associated with high mortality (27%) and low THA conversion rates (44%). Among those with THA conversion, 75% required revisions primarily due to infections related to continued drug use. Due to the high risk of infection, THA conversion is not recommended for patients with ongoing drug use. The use of an antibiotic spacer at the time of GRA was associated with improved infection resolution after a single admission. Larger prospective studies are necessary to determine the optimal management of these patients.

## Figures and Tables

**Figure 1 antibiotics-13-00782-f001:**
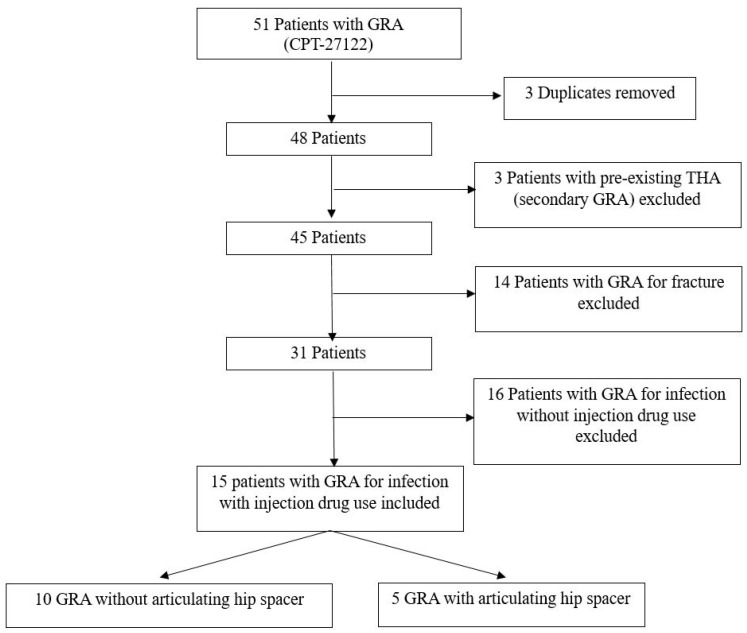
Flowchart demonstrating numbers of patients excluded and their reasons. GRA: Girdlestone resection arthroplasty. THA: total hip arthroplasty.

**Figure 2 antibiotics-13-00782-f002:**
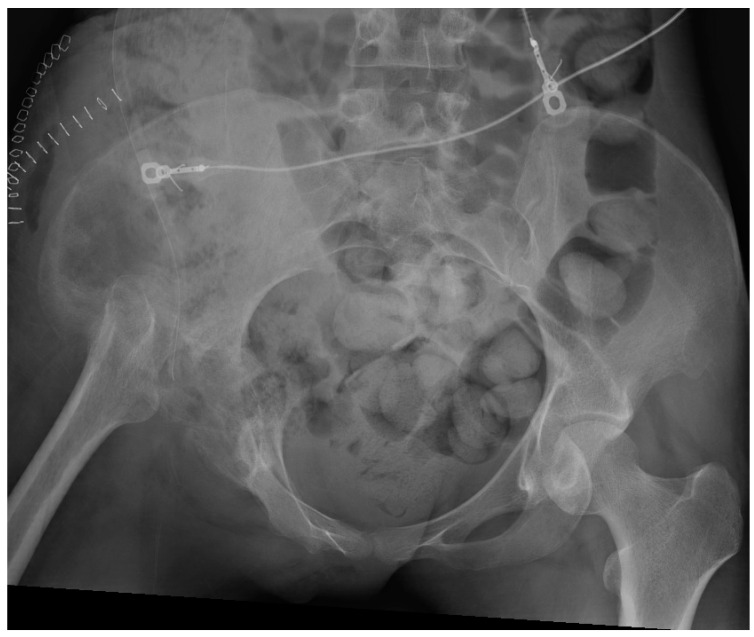
Postoperative antero-posterior pelvis radiographs of a patient who underwent a right GRA without an antibiotic spacer. Notably, the patient received a posterior approach for the GRA and a lateral window approach to debride the quadrilateral plate as the hip infection extended into the pelvis.

**Figure 3 antibiotics-13-00782-f003:**
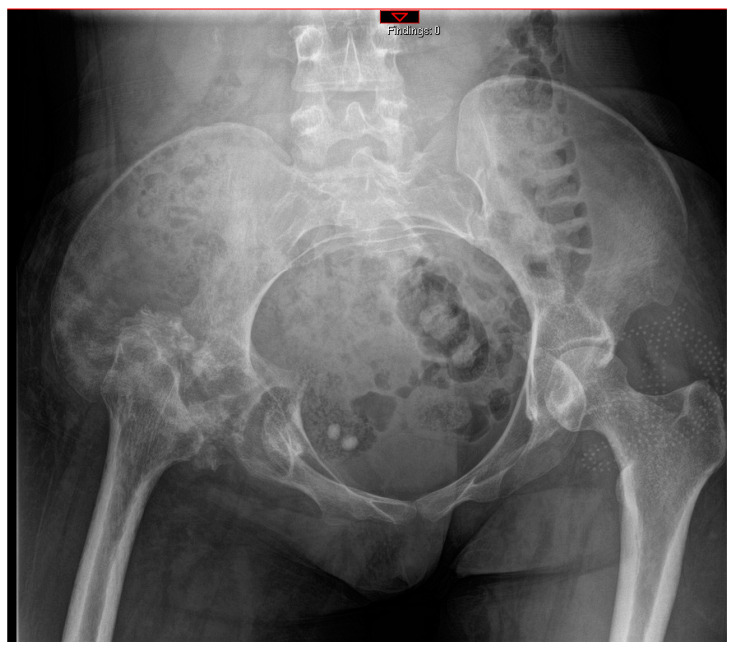
Antero-posterior pelvis radiographs of the same patient at 2 months follow-up.

**Figure 4 antibiotics-13-00782-f004:**
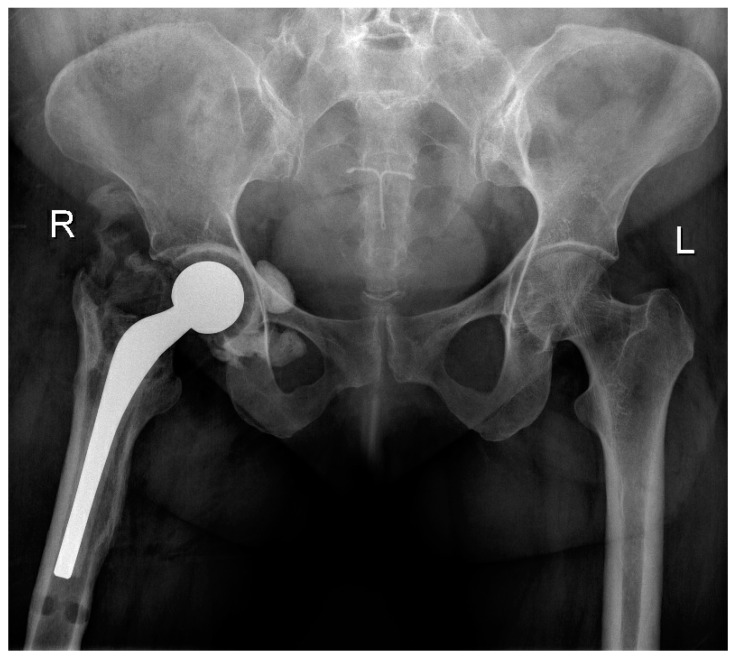
Antero-posterior pelvis radiographs of a patient who underwent a GRA with an articulating antibiotic cement spacer (Prostalac, DePuy Synthes, Raynham, MA, USA).

**Table 1 antibiotics-13-00782-t001:** Demographics and characteristics of patients with and without antibiotic spacers.

	Overall N = 15	No Antibiotic Spacer N = 10	With Antibiotic Spacer N = 5	*p*-Value
Age (years)	44 ± 11	43 ± 11	47.2 ± 11	0.459
BMI	23.1 ± 5.3	21.3 ± 4.7	26.5 ± 5.1	0.070
Gender (female)	9 (60%)	6 (60%)	3 (60%)	1.000
Active injection drug use at time of initial hospitalization	13 (87%)	10 (100%)	3 (60%)	0.095
Chronic HCV infection	9 (60%)	5 (50%)	4 (80%)	0.580
HIV infection	3 (20%)	1 (10%)	2 (40%)	0.242
History of endocarditis	3 (20%)	2 (20%)	1 (20%)	1.000
Active tobacco use	15 (100%)	10 (100%)	5 (100%)	1.000
Met SIRS criteria on presentation	4 (27%)	3 (30%)	1 (20%)	1.000
Stable housing	7 (47%)	4 (40%)	3 (60%)	0.608
Other diagnosed psychiatric disorder, other than SUD	6 (40%)	6 (60%)	0 (0%)	**0.044**
Paraplegic	2 (13%)	2 (20%)	0 (0%)	0.524
Inpatient length of stay (days)	9 (9–24)	20 (11–46)	13 (9–16)	0.297
Overall follow-up (months)	25 ± 20	24 ± 22.4	27 ± 17	0.765
Follow-up excluding deceased (months)	32 ± 20	32 ± 24	31 ± 17	0.960
Mortality	4 (27%)	3 (30%)	1 (20%)	1.000
Time to mortality (days)	78 (60–227)	70 (50–85)	369 [369]	0.180

BMI: body mass index. HCV: hepatitis C virus. HIV: human immunodeficiency virus. SIRS: systemic inflammatory response syndrome. Criteria per Sepsis-3 consensus definitions. SUD: substance use disorder. Continuous values are reported as mean ± standard deviation or median [interquartile range] based on normality as determined by the Shapiro–Wilk test. *p*-values for continuous variables were determined by the Mann–Whitney U-test or *t*-tests with equal or unequal variance as appropriate. Categorical variables are reported as N (Percent). *p*-values for categorical variables were determined by Fisher’s exact test [20]. Bold-face indicates statistical significance.

**Table 2 antibiotics-13-00782-t002:** Objective outcomes of patients with and without antibiotic spacers.

	Overall N = 9	No Antibiotic Spacer N = 5	With Antibiotic Spacer N = 4	*p*-Value
Conversion to THA	4 (44%)	1 (20%)	3 (75%)	0.206
THA requiring revision	3/4 (75%)	1/4 (25%)	2/4 (50%)	1.000
Able to ambulate at final follow-up	7 (78%)	3 (60%)	4 (100%)	0.444
Active intravenous drug use at final follow-up	3 (33%)	2 (40%)	1 (25%)	1.000
Resolution of infection following initial procedure and hospitalization	3 (33%)	0 (0%)	3 (75%)	**0.048**
Completed antibiotic course as prescribed	5 (56%)	2 (40%)	3 (75%)	0.524
Second admission for infection in ipsilateral hip	6 (67%)	5 (100%)	1 (25%)	**0.048**
Complete resolution of hip infection at final follow-up	6 (67%)	3 (60%)	3 (75%)	1.000

Patients who died or were paraplegic were excluded from these analyses. Values are reported as N (Percent). *p*-values for categorical variables were determined by Fisher’s exact test. Bold-face indicates statistical significance.

**Table 3 antibiotics-13-00782-t003:** Patient-reported outcome measures.

	Overall N = 4	Patient 3 (No Spacer)	Patient 4 (No Spacer)	Patient 13 (Spacer)	Patient 14 (Spacer)
EQ-5D mobility	3.3 ± 1.3	5	2	3	3
EQ-5D self-care	1.3 ± 0.5	2	1	1	1
EQ-5D activity	2.3 ± 1.5	4	1	3	1
EQ-5D pain	3.0 ± 1.4	5	2	3	2
EQ-5D anxiety	1.8 ± 1.0	3	1	1	2
NRS least pain	3.0 ± 2.5	5	2	5	0
NRS most pain	6.8 ± 2.5	8	8	8	3
NRS average pain	3.8 ± 3.5	8	2	5	0
Oxford Hip Score	30 ± 13	16	34	24	46

Values are reported as mean ± standard deviation. In total, 4/9 (44%) eligible patients completed patient-reported outcome measures (PROMs). Patients who were deceased or paraplegic were not considered eligible for PROM collection.

**Table 4 antibiotics-13-00782-t004:** Inpatient psychiatric intervention with and without antibiotic spacers.

	Overall N = 15	No Antibiotic Spacer N = 10	With Antibiotic Spacer N = 5	*p*-Value
SUD team consult	12 (80%)	9 (90%)	3 (60%)	0.242
Agreed to inpatient SUD treatment	8 (53%)	6 (60%)	2 (40%)	0.608
Psychiatric consult	8 (53%)	8 (80%)	0 (0%)	**0.007**
Peer recovery consult	8 (53%)	7 (70%)	1 (20%)	0.119

SUD: substance use disorder. Categorical variables are reported as N (Percent). *p*-values for categorical variables were determined by Fisher’s exact test. Bold-face indicates statistical significance.

## Data Availability

The datasets presented in this article are not readily available because the data contain protected health information that is protected under the United States Health Insurance Portability and Accountability Act. Requests to access the datasets should be directed to Henry Tout Shu.

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
