# Peer review of "Poor Outcomes of Girdlestone Resection Arthroplasty in Injection Drug Users: A Retrospective Study"

_antibiotics, 2024, doi:10.3390/antibiotics13080782_

Round 1

Reviewer 1 Report

Comments and Suggestions for Authors

For Title and Abstract:

Title Improvement:

The current title is too long and complex, making it difficult to quickly grasp the main focus. A more concise title would be preferable.

Example: "Poor Outcomes of Girdlestone Resection Arthroplasty in Injection Drug Users: A Retrospective Study."

Abstract Clarity:

The abstract should be clear and structured, starting with the background, methods, results, and conclusion. This structure is somewhat followed but could be improved for clarity and flow.

Example: "This retrospective cohort study investigates the outcomes of Girdlestone resection arthroplasty (GRA) in injection drug users with septic hip arthritis."

Background and Purpose:

The purpose of the study should be stated more clearly and concisely at the beginning of the abstract.

Example: "The study aims to evaluate the clinical outcomes of primary Girdlestone resection arthroplasty (GRA) in patients with septic hip arthritis related to injection drug use."

Methodology Description:

The methodology lacks detail regarding the patient selection criteria and the specific methods used for data collection and analysis.

Example: "Patients who underwent primary GRA for septic hip arthritis secondary to injection drug use at two academic trauma centers from 2015 to 2023 were retrospectively reviewed. Data on patient demographics, surgical details, and follow-up outcomes were collected and analyzed."

Data Presentation:

The presentation of numerical data should be more precise, avoiding unnecessary repetition and ensuring all figures are accurate.

Example: "The cohort included 15 patients, with a mean age of 44 ± 11 years and a mean follow-up period of 25 ± 20 months."

Results Section:

The results should be presented in a clear and logical order, emphasizing the key findings without redundancy.

Example: "Among the 15 patients, overall mortality was 27%, and only 4 patients underwent total hip arthroplasty (THA) following GRA. Infection resolution rates were significantly higher in patients who received an antibiotic spacer (75% vs. 0%, P=0.048)."

Conclusion and Implications:

The conclusion should succinctly summarize the study's main findings and their implications for clinical practice.

Example: "GRA in injection drug users is associated with high mortality and low conversion rates to THA. The use of an antibiotic spacer during GRA significantly improves infection resolution rates. Larger studies are required to determine the optimal management strategies for this patient population."

Language and Terminology:

The language should be formal and precise, avoiding colloquial expressions and ensuring the use of appropriate medical terminology.

Example: Replace "paucity of literature" with "limited literature" and "poor outcomes" with "suboptimal clinical outcomes."

Grammar and Syntax:

Attention to grammatical accuracy and syntactic variety will enhance the readability and professionalism of the text.

Example: "This retrospective cohort study addresses the gap in literature concerning Girdlestone resection arthroplasty outcomes in injection drug users with septic hip arthritis."

Keywords Selection:

Keywords should be selected to maximize the visibility of the article in academic searches.

Example: "Keywords: Girdlestone resection arthroplasty, injection drug use, septic hip arthritis, antibiotic spacer, clinical outcomes."

For Introduction:

Clarity and Precision:

The introduction should provide clear and precise information without unnecessary repetition or ambiguity. The use of terms such as "Girdlestone procedure/situation" is redundant and should be avoided.

Example: "The Girdlestone resection arthroplasty (GRA), an eponymous term for the resection of the femoral head and neck, was first described by Gathorne Robert Girdlestone in the early 20th century for treating hips infected by tuberculosis."

Historical Context:

The historical context of the GRA should be concisely mentioned and then move on to its current applications more directly.

Example: "Initially described for tuberculosis-infected hips, GRA is now primarily performed in elderly patients with infected total hip arthroplasty (THA) who are unsuitable for revision THA."

Target Population:

The introduction should clearly define the target population and the rationale for choosing GRA over THA without convoluted sentences.

Example: "Younger patients with hip osteomyelitis, particularly those with intravenous drug use, often require GRA due to severe bony lysis and high infection risk, making THA impractical."

Risk Factors and Rationale:

The explanation of risk factors and rationale for GRA should be succinct and logically structured.

Example: "Intravenous drug users are at high risk for hip osteomyelitis due to hematogenous bacterial seeding. Performing GRA provides definitive treatment despite the high relapse rates and severe bony lysis, which complicate THA."

Challenges and Considerations:

The challenges and considerations of performing THA in these patients should be clearly articulated, emphasizing the complexity of their clinical management.

Example: "THA in these patients is fraught with challenges, including elevated implant infection risk and the high likelihood of relapse, even after extended abstinence periods."

Current Trends and Importance:

The introduction should emphasize the current trends in injection drug use and its implications for orthopedic practice, underlining the study's relevance.

Example: "With rising injection drug use rates globally, orthopedic surgeons increasingly encounter septic hips in this demographic, necessitating a thorough understanding of GRA outcomes."

Literature Gap:

The gap in the existing literature should be highlighted more effectively, stressing the need for this study.

Example: "Despite the long history of GRA, the quality of life and functional outcomes post-GRA in younger injection drug users remain poorly characterized. Existing studies predominantly address secondary procedures for prosthetic joint infections, with limited data on primary GRA outcomes in this specific population."

Study Purpose and Hypotheses:

The study's purpose and hypotheses should be clearly and concisely stated towards the end of the introduction, without repetitive language.

Example: "This study aims to describe the outcomes of primary GRA for hip osteomyelitis in injection drug users and compare outcomes between those who received an antibiotic spacer and those who did not. We hypothesize that antibiotic spacer use correlates with better patient-reported outcome measures (PROMs) and that overall, there will be a low conversion rate to THA with a high revision rate for those who did receive THAs."

Grammar and Syntax:

Ensure grammatical accuracy and syntactic variety to enhance readability and professionalism.

Example: "As the incidence of injection drug use rises, orthopedic traumatologists are increasingly likely to treat septic hips secondary to this cause. However, comprehensive data on the outcomes of GRA in these patients are scarce."

For Results:

Overall Cohort Demographics and Characteristics:

The section needs better organization and clarity to enhance readability. The flow of information is currently cluttered and difficult to follow.

Example: Break down the information into clearly defined sub-sections such as patient demographics, infection characteristics, and treatment details. This would help in presenting the data in a more structured manner.

Clarity and Precision in Data Presentation:

The description of inclusion criteria and the characteristics of the cohort should be precise and succinct. Avoid redundancy and ensure that the criteria are clearly stated.

Example: "Out of 51 patients who underwent GRA at two institutions, 15 (29%) had primary GRA for infection secondary to injection drug use. Inclusion was based on radiographic or MRI signs of infection, elevated inflammatory markers (CRP, ESR, CBC), presence of purulence or infection in operative reports, and/or positive intraoperative cultures."

Detailed Descriptions:

The details provided for specific statistics are scattered and need to be centralized for better understanding. Also, avoid unnecessary repetition of information.

Example: "Of the 15 patients, 10 (67%) did not receive an antibiotic spacer at the time of GRA. The mean age was 44±11 years, BMI was 23.1±5.3, and follow-up duration was 25±20 months (32±20 months excluding deceased patients)."

Statistical Comparisons:

Ensure statistical comparisons are clearly presented and explained, emphasizing significant findings.

Example: "Patients who did not receive an antibiotic spacer had a significantly higher prevalence of previously diagnosed psychiatric conditions (60% vs. 0%, P=0.044)."

Mortality and Other Significant Outcomes:

Present mortality data and other significant outcomes in a clear, concise manner.

Example: "Overall mortality was 27% (4/15), with a median time to death of 78 [60-227] days. Mortality was higher in the non-antibiotic spacer group (30% vs. 20%, P=0.180)."

Infection and Microbiology Data:

This section is detailed but could benefit from a more structured presentation, separating data on blood cultures, hip aspirations, and intraoperative cultures.

Example: "Among the 15 patients, 4 (27%) met SIRS criteria on presentation. Positive blood cultures were found in 4 patients, primarily growing MSSA and GAS, consistent with intraoperative cultures."

Objective Outcomes:

This section should provide a clear comparison between the groups and highlight significant differences or notable trends.

Example: "Following GRA, 4 (44%) patients received a THA, of which 3 (75%) required revisions. Among the 9 patients analyzed for outcomes, 7 (78%) were ambulatory, and 6 (67%) had complete resolution of their infection."

Patient-Reported Outcome Measures (PROMs):

The presentation of PROMs is limited and lacks depth. Ensure all relevant data is included and clearly presented.

Example: "Of the 9 eligible patients, only 4 (44%) completed PROMs, showing no significant differences between those with and without antibiotic spacers. The highest difficulty was reported in the EQ-5D Mobility dimension (mean 3.3±1.3)."

In-hospital Psychiatric Interventions:

This section could be streamlined to better reflect the psychiatric interventions received and their distribution among patient groups.

Example: "During hospitalization, 80% (12/15) of patients were consulted by the SUD team, with 67% agreeing to inpatient rehabilitation. Psychiatric consults were significantly more common in the non-antibiotic spacer group (53% vs. 0%, P=0.007)."

Grammar and Syntax:

Ensure grammatical accuracy and syntactic variety to enhance readability and professionalism.

Example: "Patients who underwent GRA without an antibiotic spacer had a significantly higher rate of previously diagnosed psychiatric conditions compared to those who received a spacer."

For Discussion:

Clarity and Focus:

The discussion should be concise and focused, avoiding unnecessary repetition. Ensure that each point is clearly articulated.

Example: The repeated mention of high mortality rates and low THA conversion rates can be condensed into a single, clear statement at the beginning.

Interpretation of Results:

The interpretation of the results needs to be more direct and analytical, emphasizing the significance of the findings without ambiguity.

Example: "The study found a high postoperative mortality rate (27%) and a low conversion rate to THA (44%), with a significant proportion of THA conversions requiring revisions due to infection."

Association vs. Causation:

Clearly distinguish between association and causation, especially when discussing the impact of antibiotic spacers.

Example: "While the use of an antibiotic spacer was associated with higher infection resolution rates, this may reflect the selection of patients with less severe disease rather than the effect of the spacer itself."

Comparison with Existing Literature:

The comparison with existing literature should be comprehensive and well-integrated, not limited to a single case report.

Example: "Limited literature exists on GRA outcomes in injection drug users, with Maguire et al. reporting a case without long-term outcome data. Other studies on primary GRA in infection settings have shown similar mortality and functional outcomes."

Detailed Analysis of Findings:

Provide a deeper analysis of specific findings, such as the types of infectious organisms and their implications.

Example: "The predominance of skin flora infections, consistent with previous studies, underscores the need for targeted antibiotic therapy. Notably, polymicrobial infections in paraplegic patients suggest a need for vigilance regarding opportunistic infections from the gastrointestinal tract."

Clinical Implications and Recommendations:

The discussion should provide clear clinical implications and recommendations based on the findings.

Example: "Given the high relapse rates and complex infection profiles, multidisciplinary management involving infectious disease specialists, psychiatrists, and substance use disorder experts is crucial for improving outcomes."

Grammar and Syntax:

Ensure grammatical accuracy and syntactic variety to enhance readability and professionalism.

Example: "The study highlights the need for improved management strategies for injection drug users with septic hip arthritis, emphasizing the importance of comprehensive, multidisciplinary care."

For Limitations:

Sample Size:

The discussion of the small sample size should be concise and focus on its impact on the study's generalizability and statistical power.

Example: "The small sample size is a significant limitation, impacting the generalizability and statistical power of the study's findings."

Retrospective Nature:

The retrospective design's limitations should be clearly outlined, emphasizing the issues of selection bias and the inability to establish causality.

Example: "The retrospective nature of the study introduces selection bias and limits the ability to establish causality regarding the efficacy of antibiotic spacers in facilitating single-stage GRA."

Selection Bias:

The issue of selection bias should be explicitly detailed, including how it affects the study’s conclusions about antibiotic spacers.

Example: "Selection bias is evident as patients who did not receive antibiotic spacers were more likely to have psychiatric diagnoses, potentially skewing the outcomes."

Feasibility of Antibiotic Spacers:

Clearly explain why antibiotic spacers might not be feasible in severe osteomyelitis cases, focusing on the technical and clinical constraints.

Example: "In cases of severe osteomyelitis resulting in extensive bony loss, the placement of antibiotic spacers may not be feasible due to insufficient structural integrity of the acetabulum and proximal femur post-debridement."

Patient-Reported Outcome Measures (PROMs):

The difficulty in collecting PROMs should be highlighted, including specific challenges related to the patient population’s instability and lack of reliable contact information.

Example: "Collecting PROMs in this population is challenging due to instability in housing and unreliable contact information, resulting in only 4 patients completing the surveys."

Follow-up Challenges:

Address the challenges in conducting in-office follow-ups and how this impacts the ability to perform THA post-GRA.

Example: "In-office follow-ups are difficult, reducing the likelihood of patients receiving a THA following GRA due to the transient nature of the patient population and their instability."

Overall Impact of Limitations:

Summarize how these limitations collectively affect the study's conclusions and the need for further research.

Example: "These limitations, including the small sample size, retrospective design, and follow-up challenges, underscore the need for larger, prospective studies to better evaluate the outcomes of GRA in injection drug users."

For Methods:

Study Approval and Design:

The mention of Institutional Review Board (IRB) approval is appropriate, but the information should be condensed and presented more clearly.

Example: "Following Institutional Review Board approval (JHM IRB00391703), a retrospective study was conducted at two U.S. academic trauma centers from January 2015 to December 2023."

Patient Identification and Selection:

The process of identifying and selecting patients should be detailed succinctly, ensuring clarity and completeness.

Example: "Patients who underwent GRA (CPT-27122) were identified through electronic medical records. Inclusion criteria were primary GRA for infected hip secondary to injection drug use; patients with secondary GRA or preexisting THA were excluded."

Surgical Technique Description:

The surgical technique section is verbose and could benefit from being more concise and focused.

Example: "All patients received a posterior hip approach, with a lateral window approach used if infection extended into the pelvis. The decision to use an antibiotic spacer was made by the attending surgeon based on bone integrity post-debridement."

Figures and Examples:

The reference to figures within the text can be distracting and is better placed in figure legends or separate explanations.

Example: "Figures 1 and 2 illustrate cases where severe bony involvement precluded spacer placement, while Figure 3 shows a patient with an antibiotic hip spacer."

Objective Outcome Measures:

The description of objective outcome measures should be clear and organized, with a focus on key outcomes.

Example: "Primary outcomes included mortality, conversion to THA, revision THA, ambulation status, and infection resolution. Secondary outcomes included patient demographics, social history, and medical comorbidities."

Patient Demographics and Comorbidities:

The description of demographics and comorbidities should be precise and focused, avoiding repetition.

Example: "Demographics collected included age, BMI, and gender. Comorbidities included chronic HCV, history of endocarditis, and psychiatric disorders."

PROMs Collection:

The description of PROMs should be detailed but concise, explaining the collection process and tools used.

Example: "PROMs were collected via phone-administered surveys with verbal consent. Measures included the EQ-5D-5L, NRS for pain, and Oxford Hip Score, with paraplegic and deceased patients excluded from collection."

Statistical Analysis:

The statistical methods should be described clearly, with an emphasis on the tests used and the rationale for their selection.

Example: "Continuous variables were assessed for normality using Shapiro-Wilks tests. Comparisons were made using Mann-Whitney tests for non-normal distributions or T-tests for normal distributions. Categorical variables were compared using Fisher’s exact test, with an α value of 0.05."

For Conclusions:

Conciseness and Clarity:

The conclusions should be succinct and clear, summarizing the key findings and their implications without unnecessary detail.

Example: "GRA in patients with septic hip arthritis secondary to injection drug use is associated with high mortality (27%) and low THA conversion rates (44%)."

Emphasis on Key Findings:

Ensure the emphasis is placed on the most significant findings and their implications for clinical practice.

Example: "Among those who underwent THA, 75% required revisions due to infections related to continued drug use."

Clinical Recommendations:

Recommendations should be clear and based on the findings, avoiding overly strong or ambiguous statements.

Example: "Due to the high risk of infection, second-stage total hip replacement is not recommended for patients with ongoing drug use."

Impact of Antibiotic Spacer:

The impact of using an antibiotic spacer should be clearly stated, emphasizing its association with infection resolution.

Example: "The use of an antibiotic spacer at the time of GRA was associated with improved infection resolution rates after a single admission."

Call for Further Research:

The need for further research should be highlighted, specifying the type of studies required.

Example: "Further large-scale, prospective studies are necessary to determine the optimal management strategies for this patient population."

Grammatical Precision:

Ensure grammatical accuracy and professional tone.

Example: "Further larger-scale studies" can be rephrased to "Further large-scale studies" for better readability.

Comments on the Quality of English Language

For the section "Comments on the Quality of English Language (will be shown to authors)," all elements have already been explained in the section above.

Author Response

Dear Reviewer 1,

Thank you for reviewing our manuscript and the thoughtful comments. We believe these comments have greatly improved the clarity of our manuscript. Our responses to the comments are below. We have made major revisions to almost every paragraph, and thus have not highlighted all the changed text. However, all the changes are tracked in the manuscript. We hope that you find our responses satisfactory.

Reviewer 1 Comments

For Title and Abstract:

Title Improvement:

The current title is too long and complex, making it difficult to quickly grasp the main focus. A more concise title would be preferable.

Example: "Poor Outcomes of Girdlestone Resection Arthroplasty in Injection Drug Users: A Retrospective Study."

We have now revised the title to:

"Poor Outcomes of Girdlestone Resection Arthroplasty in Injection Drug Users: A Retrospective Study."

Abstract Clarity:

The abstract should be clear and structured, starting with the background, methods, results, and conclusion. This structure is somewhat followed but could be improved for clarity and flow.

Example: "This retrospective cohort study investigates the outcomes of Girdlestone resection arthroplasty (GRA) in injection drug users with septic hip arthritis."

We have now revised the start of the abstract to be more concise to:

“This retrospective cohort study aims to investigate the clinical outcomes of Girdlestone resection arthroplasty (GRA) in injection drug users with septic hip arthritis.”

Background and Purpose:

The purpose of the study should be stated more clearly and concisely at the beginning of the abstract.

Example: "The study aims to evaluate the clinical outcomes of primary Girdlestone resection arthroplasty (GRA) in patients with septic hip arthritis related to injection drug use."

We have now revised:

“This retrospective cohort study aims to investigate the clinical outcomes of Girdlestone resection arthroplasty (GRA) in injection drug users with septic hip arthritis.”

Methodology Description:

The methodology lacks detail regarding the patient selection criteria and the specific methods used for data collection and analysis.

Example: "Patients who underwent primary GRA for septic hip arthritis secondary to injection drug use at two academic trauma centers from 2015 to 2023 were retrospectively reviewed. Data on patient demographics, surgical details, and follow-up outcomes were collected and analyzed."

We have now revised this to state:

“Patients who underwent primary GRA for septic hip arthritis secondary to injection drug use at two academic trauma centers from 2015 to 2023 were retrospectively reviewed. Patient demographics, surgical details, and follow-up outcomes, including patient-reported outcome measures, were collected and analyzed.”

Data Presentation:

The presentation of numerical data should be more precise, avoiding unnecessary repetition and ensuring all figures are accurate.

Example: "The cohort included 15 patients, with a mean age of 44 ± 11 years and a mean follow-up period of 25 ± 20 months."

We have now revised this to:

“The cohort included 15 patients, with a mean age of 44 ± 11 years and a mean follow-up period of 25 ± 20 months”

Results Section:

The results should be presented in a clear and logical order, emphasizing the key findings without redundancy.

Example: "Among the 15 patients, overall mortality was 27%, and only 4 patients underwent total hip arthroplasty (THA) following GRA. Infection resolution rates were significantly higher in patients who received an antibiotic spacer (75% vs. 0%, P=0.048)."

We have now revised the results to state:

“Among the 15 patients, overall mortality was 27%, and only 4 patients underwent total hip arthroplasty (THA) following GRA. Infection resolution rates were significantly higher in patients who received an antibiotic spacer (75% vs. 0%, P=0.048).”

Conclusion and Implications:

The conclusion should succinctly summarize the study's main findings and their implications for clinical practice.

Example: "GRA in injection drug users is associated with high mortality and low conversion rates to THA. The use of an antibiotic spacer during GRA significantly improves infection resolution rates. Larger studies are required to determine the optimal management strategies for this patient population."

Have now revised this to state: “GRA in injection drug users is associated with high mortality and low conversion rates to THA. The use of an antibiotic spacer during GRA significantly improves infection resolution rates. Larger studies are required to determine the optimal management strategies for this patient population.”

Language and Terminology:

The language should be formal and precise, avoiding colloquial expressions and ensuring the use of appropriate medical terminology.

Example: Replace "paucity of literature" with "limited literature" and "poor outcomes" with "suboptimal clinical outcomes."

We have revised this now throughout the manuscript.

Grammar and Syntax:

Attention to grammatical accuracy and syntactic variety will enhance the readability and professionalism of the text.

Example: "This retrospective cohort study addresses the gap in literature concerning Girdlestone resection arthroplasty outcomes in injection drug users with septic hip arthritis."

We have revised this now throughout the manuscript.

Keywords Selection:

Keywords should be selected to maximize the visibility of the article in academic searches.

Example: "Keywords: Girdlestone resection arthroplasty, injection drug use, septic hip arthritis, antibiotic spacer, clinical outcomes."

Have now revised Keywords to the above.

For Introduction:

Clarity and Precision:

The introduction should provide clear and precise information without unnecessary repetition or ambiguity. The use of terms such as "Girdlestone procedure/situation" is redundant and should be avoided.

Example: "The Girdlestone resection arthroplasty (GRA), an eponymous term for the resection of the femoral head and neck, was first described by Gathorne Robert Girdlestone in the early 20th century for treating hips infected by tuberculosis."

Historical Context:

The historical context of the GRA should be concisely mentioned and then move on to its current applications more directly.

Example: "Initially described for tuberculosis-infected hips, GRA is now primarily performed in elderly patients with infected total hip arthroplasty (THA) who are unsuitable for revision THA."

Have now revised the first 2 sentences to be more concise:

“The Girdlestone resection arthroplasty (GRA) is an eponymous term for resection of the femoral head and neck. While initially described by Gathorne Robert Girdlestone in the early 20th century for tuberculosis infected hips, GRA is now primarily performed for elderly patients with infected total hip arthroplasty (THA) who are unable to receive a revision THA.[1-5]”

Target Population:

The introduction should clearly define the target population and the rationale for choosing GRA over THA without convoluted sentences.

Example: "Younger patients with hip osteomyelitis, particularly those with intravenous drug use, often require GRA due to severe bony lysis and high infection risk, making THA impractical."

Risk Factors and Rationale:

The explanation of risk factors and rationale for GRA should be succinct and logically structured.

Example: "Intravenous drug users are at high risk for hip osteomyelitis due to hematogenous bacterial seeding. Performing GRA provides definitive treatment despite the high relapse rates and severe bony lysis, which complicate THA."

Challenges and Considerations:

The challenges and considerations of performing THA in these patients should be clearly articulated, emphasizing the complexity of their clinical management.

Example: "THA in these patients is fraught with challenges, including elevated implant infection risk and the high likelihood of relapse, even after extended abstinence periods."

Have revised the second paragraph in the introduction to be much more concise:

“One such population that is at risk for hip osteomyelitis is patients with injection drug use, where hematogenous seeding of bacteria may lead to osteomyelitis of the hip.[6,7] These patients often require GRA due to severe bony lysis, high re-infection risk, and high rates of drug relapse, making THA impractical.[6-11] Moreover, as these patients are generally younger than the typical patients undergoing THA, it is important to under-stand the long-term implications of performing GRA in these patients.”

Current Trends and Importance:

The introduction should emphasize the current trends in injection drug use and its implications for orthopedic practice, underlining the study's relevance.

Example: "With rising injection drug use rates globally, orthopedic surgeons increasingly encounter septic hips in this demographic, necessitating a thorough understanding of GRA outcomes."

Literature Gap:

The gap in the existing literature should be highlighted more effectively, stressing the need for this study.

Example: "Despite the long history of GRA, the quality of life and functional outcomes post-GRA in younger injection drug users remain poorly characterized. Existing studies predominantly address secondary procedures for prosthetic joint infections, with limited data on primary GRA outcomes in this specific population."

Study Purpose and Hypotheses:

The study's purpose and hypotheses should be clearly and concisely stated towards the end of the introduction, without repetitive language.

Example: "This study aims to describe the outcomes of primary GRA for hip osteomyelitis in injection drug users and compare outcomes between those who received an antibiotic spacer and those who did not. We hypothesize that antibiotic spacer use correlates with better patient-reported outcome measures (PROMs) and that overall, there will be a low conversion rate to THA with a high revision rate for those who did receive THAs."

Grammar and Syntax:

Ensure grammatical accuracy and syntactic variety to enhance readability and professionalism.

Example: "As the incidence of injection drug use rises, orthopedic traumatologists are increasingly likely to treat septic hips secondary to this cause. However, comprehensive data on the outcomes of GRA in these patients are scarce."

We have now revised both these paragraphs to be far more concise, in accordance with the suggestions above.

“With rising injection drug use rates globally, orthopedic surgeons are more likely to encounter patients with septic hips in this demographic, necessitating a thorough under-standing of GRA outcomes.[12-15] Despite the long history of the GRA, the quality of life and functional outcomes post-GRA in younger patients have been poorly characterized.[3,7,16-18] Existing studies predominantly address secondary procedures for pros-thetic joint infections, with limited data on primary GRA outcomes in this specific population.[3,7,16-18]

Therefore, this study aims to describe the outcomes of primary GRA for hip osteomyelitis in injection drug users and compare outcomes between those who received an antibiotic spacer and those who did not. We hypothesize that antibiotic spacer use correlates with better patient-reported outcome measures (PROMs) and that overall, there will be a low conversion rate to THA with a high revision rate for those who did receive THAs.”

For Results:

Overall Cohort Demographics and Characteristics:

The section needs better organization and clarity to enhance readability. The flow of information is currently cluttered and difficult to follow.

Example: Break down the information into clearly defined sub-sections such as patient demographics, infection characteristics, and treatment details. This would help in presenting the data in a more structured manner.

Clarity and Precision in Data Presentation:

The description of inclusion criteria and the characteristics of the cohort should be precise and succinct. Avoid redundancy and ensure that the criteria are clearly stated.

Example: "Out of 51 patients who underwent GRA at two institutions, 15 (29%) had primary GRA for infection secondary to injection drug use. Inclusion was based on radiographic or MRI signs of infection, elevated inflammatory markers (CRP, ESR, CBC), presence of purulence or infection in operative reports, and/or positive intraoperative cultures."

Detailed Descriptions:

The details provided for specific statistics are scattered and need to be centralized for better understanding. Also, avoid unnecessary repetition of information.

Example: "Of the 15 patients, 10 (67%) did not receive an antibiotic spacer at the time of GRA. The mean age was 44±11 years, BMI was 23.1±5.3, and follow-up duration was 25±20 months (32±20 months excluding deceased patients)."

Statistical Comparisons:

Ensure statistical comparisons are clearly presented and explained, emphasizing significant findings.

Example: "Patients who did not receive an antibiotic spacer had a significantly higher prevalence of previously diagnosed psychiatric conditions (60% vs. 0%, P=0.044)."

Have now revised the first part of the results section to be more concise and clearer with the recommendations above:

“Out of 51 patients who received a GRA (CPT-27122) at the authors’ 2 institutions, 15 (29%) received a primary GRA for infection secondary to injection drug use. Inclusion was based on the documentation of injection drug use in patient’s history and radiographic or MRI signs of infection, elevated inflammatory markers [C-reactive protein (CRP), erythro-cyte sedimentation rate (ESR), complete blood count (CBC)], presence of purulence or in-fection in operative reports, and/or positive intraoperative cultures. Of the 15 patients, 10 (67%) did not receive an antibiotic spacer at the time of GRA. The mean age was 44±11 years and BMI was 23.1±5.3. The mean follow-up was 25±20 months (32±20 months when excluding deceased patients). When comparing the demographics and characteristics of patients who received an antibiotic spacer versus those who did not, the only significant factor was the presence of previously diagnosed psychiatric conditions (Table 1). No pa-tients had Hepatitis B infection at time of GRA. Patients who did not receive an antibiotic spacer had a significantly greater percentage of previously diagnosed psychiatric condi-tions (60% versus 0%, P=0.044). Notably, all patients (15/15) had active tobacco use at the time of their primary GRA.”

Mortality and Other Significant Outcomes:

Present mortality data and other significant outcomes in a clear, concise manner.

Example: "Overall mortality was 27% (4/15), with a median time to death of 78 [60-227] days. Mortality was higher in the non-antibiotic spacer group (30% vs. 20%, P=0.180)."

Infection and Microbiology Data:

This section is detailed but could benefit from a more structured presentation, separating data on blood cultures, hip aspirations, and intraoperative cultures.

Example: "Among the 15 patients, 4 (27%) met SIRS criteria on presentation. Positive blood cultures were found in 4 patients, primarily growing MSSA and GAS, consistent with intraoperative cultures."

Have now created a new sub-section and revised it to be far more concise, as recommended.

“Sepsis and Mortality

Among the 15 patients, 4 (27%) patients met SIRS criteria per Sepsis-3 Consensus Definitions.[20] Positive blood cultures were found in 4 (27%) patients, primarily growing methicillin-sensitive Staphylococcus aureus (MSSA), and Group A Streptococcus species (GAS), all of which were consistent with subsequent intraoperative cultures. Of these 4 patients with positive blood cultures, only 1 met SIRS criteria on presentation. No septic patients or patients with positive blood cultures had seeding of other joints during their admission for GRA.

Overall mortality was 27% (4/15), with a median time to death of 78 [60-227] days. Three patients who did not receive antibiotic spacers died, whereas 1 patient who received an antibiotic spacer died (Table 1, P=0.180). Two (13%) patients were paraplegic, both of whom did not receive an antibiotic spacer. Deidentified descriptions of each patient are available in Supplemental Table 1. Regarding the deaths, 1 patient died from a subse-quent opioid overdose, 1 patient died from massive PE 3 months after GRA, and 2 patients had unknown causes of death.”

Have now also separated the microbiology regarding hip aspirations and intraoperative cultures.

Objective Outcomes:

This section should provide a clear comparison between the groups and highlight significant differences or notable trends.

Example: "Following GRA, 4 (44%) patients received a THA, of which 3 (75%) required revisions. Among the 9 patients analyzed for outcomes, 7 (78%) were ambulatory, and 6 (67%) had complete resolution of their infection."

Have now revised and reformatted this paragraph to be more concise.

“Following GRA, 4 patients received a THA ,of which 3 (75%) required revisions. Among the 9 patients analyzed for outcomes, 7 (78%) were ambulatory and 6 (67%) had complete resolution of their hip infection (Table 2). Of the 3 revision THAs, 2 were due to septic loosening, or infected THA in the setting of continued injection drug use. These 2 infected THAs were treated with an explant and placement of a subsequent antibiotic spacer. Unfortunately, 1 of those 2 patients fractured their femur below the antibiotic spacer, and had subsequent open reduction and internal fixation, nonunion, and ultimately resection of the proximal femur with removal of all implants. The one revision not due to infection was caused by recurrent dislocations and was successfully treated with a revision of the acetabular cup to a larger size.”

Patient-Reported Outcome Measures (PROMs):

The presentation of PROMs is limited and lacks depth. Ensure all relevant data is included and clearly presented.

Example: "Of the 9 eligible patients, only 4 (44%) completed PROMs, showing no significant differences between those with and without antibiotic spacers. The highest difficulty was reported in the EQ-5D Mobility dimension (mean 3.3±1.3)."

Have now revised this paragraph to be much more concise:

“Of the 9 eligible patients, only 4 (44%) patients completed PROMs, showing no sig-nificant differences between those with and without antibiotic spacers (Table 3). The EQ-5D-5L Mobility dimension was associated with the greatest difficulty (mean 3.3±1.3).”

Have also further added to limitations regarding the limited amount of PROMs we were able to collect.:

“Additionally, only 4 patients were able to complete PROM surveys, greatly limiting the applicability of these findings.”

In-hospital Psychiatric Interventions:

This section could be streamlined to better reflect the psychiatric interventions received and their distribution among patient groups.

Example: "During hospitalization, 80% (12/15) of patients were consulted by the SUD team, with 67% agreeing to inpatient rehabilitation. Psychiatric consults were significantly more common in the non-antibiotic spacer group (53% vs. 0%, P=0.007)."

Have now revised this section to be more concise, as per above:

“During hospitalization, 80% (12/15) of patients were consulted by the       substance use dis-order (SUD) team, of which 67% (8/12) agreeing to inpatient SUD rehabilitation. Psy-chaitric consults were significantly more common in the non-antibiotic spacer group (53% vs. 0%, P=0.007, Table 4).”

Grammar and Syntax:

Ensure grammatical accuracy and syntactic variety to enhance readability and professionalism.

Example: "Patients who underwent GRA without an antibiotic spacer had a significantly higher rate of previously diagnosed psychiatric conditions compared to those who received a spacer."

Have now revised this in the first paragraph of the results section.:

“Patients who underwent GRA without antibiotic spacer had significantly higher rates of previously diagnosed conditions compared to those who received a spacer (60% vs. 0%, P=0.044 Table 1).”

For Discussion:

Clarity and Focus:

The discussion should be concise and focused, avoiding unnecessary repetition. Ensure that each point is clearly articulated.

Example: The repeated mention of high mortality rates and low THA conversion rates can be condensed into a single, clear statement at the beginning.

Interpretation of Results:

The interpretation of the results needs to be more direct and analytical, emphasizing the significance of the findings without ambiguity.

Example: "The study found a high postoperative mortality rate (27%) and a low conversion rate to THA (44%), with a significant proportion of THA conversions requiring revisions due to infection."

Have now revised to:

“This study found a high postoperative mortality rate (27%) and low rate of conversion to THA (44%), with a large proportion of THA conversions requiring revision due to infection. Nevertheless, 67% of patients had complete resolution of their hip infections.”

Association vs. Causation:

Clearly distinguish between association and causation, especially when discussing the impact of antibiotic spacers.

Example: "While the use of an antibiotic spacer was associated with higher infection resolution rates, this may reflect the selection of patients with less severe disease rather than the effect of the spacer itself."

Have now revised this discussion to be far more concise in accordance with above:

“While the use of an antibiotic hip spacer was associated with a higher rate of infection resolution (75% vs. 0%, P=0.048, Table 2), this may reflect the selection of patients with less severe disease rather than the effect of the spacer itself.”

Comparison with Existing Literature:

The comparison with existing literature should be comprehensive and well-integrated, not limited to a single case report.

Example: "Limited literature exists on GRA outcomes in injection drug users, with Maguire et al. reporting a case without long-term outcome data. Other studies on primary GRA in infection settings have shown similar mortality and functional outcomes."

Have now revised to the recommendations above.

“There is very limited literature on outcomes in injection drug users who receive pri-mary GRA, with Maguire et al. reported a case without long-term data.[19] Other recent studies regarding primary GRA in the setting of infection and/or fracture have shown sim-ilar rates of mortality and limited functional outcomes as compared to this study.[17,18] The poor outcomes in the injection drug use population are likely due to high rates of sub-stance use relapse over the course of an addiction [11,21], and high rates of psychiatric and medical comorbidities in this population despite their generally young age. [2,22,23]”

Detailed Analysis of Findings:

Provide a deeper analysis of specific findings, such as the types of infectious organisms and their implications.

Example: "The predominance of skin flora infections, consistent with previous studies, underscores the need for targeted antibiotic therapy. Notably, polymicrobial infections in paraplegic patients suggest a need for vigilance regarding opportunistic infections from the gastrointestinal tract."

Have now revised to as follows:

“The predominance of skin flora, such as MSSA, MRSA, and GAS, is consistent with previous studies, underscores the need for targeted antibiotic therapy.[6,8] Notably, polymicrobial infections in paraplegic patients suggest a need for vigilance regarding opportunistic infections from the gastrointestinal tract.[24] A higher rate of opportunistic infection and unusual organisms should be considered in this patient group due to the re-current bacteremia associated with contaminated needle use, as well as the associated risk of immunocompromise related to poor general health, malnutrition and the association with HIV and Hepatitis C.”

Clinical Implications and Recommendations:

The discussion should provide clear clinical implications and recommendations based on the findings.

Example: "Given the high relapse rates and complex infection profiles, multidisciplinary management involving infectious disease specialists, psychiatrists, and substance use disorder experts is crucial for improving outcomes."

Grammar and Syntax:

Ensure grammatical accuracy and syntactic variety to enhance readability and professionalism.

Example: "The study highlights the need for improved management strategies for injection drug users with septic hip arthritis, emphasizing the importance of comprehensive, multidisciplinary care."

Have now revised to as follows:

“As the rates of injection drugs continue to dramatically increase globally, there is a need to better understand the outcomes following treatment of infection secondary to in-jection drug use. [14,15] Given the high relapse rates and complex infection profiles, mul-tidisciplinary management involving infectious disease specialists, psychiatrists, and substance use disorder experts is crucial for improving outcomes. Nevertheless, this study highlights the need for improved management strategies for injection drug users with sep-tic hip arthritis, emphasizing the importance of comprehensive, multidisciplinary care.”

For Limitations:

Sample Size:

The discussion of the small sample size should be concise and focus on its impact on the study's generalizability and statistical power.

Example: "The small sample size is a significant limitation, impacting the generalizability and statistical power of the study's findings."

Retrospective Nature:

The retrospective design's limitations should be clearly outlined, emphasizing the issues of selection bias and the inability to establish causality.

Example: "The retrospective nature of the study introduces selection bias and limits the ability to establish causality regarding the efficacy of antibiotic spacers in facilitating single-stage GRA."

Selection Bias:

The issue of selection bias should be explicitly detailed, including how it affects the study’s conclusions about antibiotic spacers.

Example: "Selection bias is evident as patients who did not receive antibiotic spacers were more likely to have psychiatric diagnoses, potentially skewing the outcomes."

Feasibility of Antibiotic Spacers:

Clearly explain why antibiotic spacers might not be feasible in severe osteomyelitis cases, focusing on the technical and clinical constraints.

Example: "In cases of severe osteomyelitis resulting in extensive bony loss, the placement of antibiotic spacers may not be feasible due to insufficient structural integrity of the acetabulum and proximal femur post-debridement."

Patient-Reported Outcome Measures (PROMs):

The difficulty in collecting PROMs should be highlighted, including specific challenges related to the patient population’s instability and lack of reliable contact information.

Example: "Collecting PROMs in this population is challenging due to instability in housing and unreliable contact information, resulting in only 4 patients completing the surveys."

Follow-up Challenges:

Address the challenges in conducting in-office follow-ups and how this impacts the ability to perform THA post-GRA.

Example: "In-office follow-ups are difficult, reducing the likelihood of patients receiving a THA following GRA due to the transient nature of the patient population and their instability."

Overall Impact of Limitations:

Summarize how these limitations collectively affect the study's conclusions and the need for further research.

Example: "These limitations, including the small sample size, retrospective design, and follow-up challenges, underscore the need for larger, prospective studies to better evaluate the outcomes of GRA in injection drug users."

Have now revised the limitations with all the recommendations above:

“The small sample size of this study is a severe limitation, impacting the generalizability and statistical power of the study’s findings. Furthermore, the retrospective nature of this study introduces selection bias and limits the ability to establish causality regarding the efficacy of antibiotic spacers in facilitating single-stage GRA. Selection bias is evident in patients who did not receive antibiotic spacers having greater rates of psychiatric diagnoses, skewing outcomes. Furthermore, in cases of severe osteomyelitis with extensive bony loss, the placement of antibiotic spacers may not be feasible due to insufficient stability of the acetabulum and proximal femur post-debridement. Additionally, collecting PROMs in this patient population is challenging due to housing instability and unreliable contact information, resulting in only 4 patients completing PROMs. This severely limits the generalizability of the PROMs in this study. In-office follow-ups are also difficult in this patient population, further reducing the likelihood of THA conversion because of patient noncompliance.”

For Methods:

Study Approval and Design:

The mention of Institutional Review Board (IRB) approval is appropriate, but the information should be condensed and presented more clearly.

Example: "Following Institutional Review Board approval (JHM IRB00391703), a retrospective study was conducted at two U.S. academic trauma centers from January 2015 to December 2023."

Patient Identification and Selection:

The process of identifying and selecting patients should be detailed succinctly, ensuring clarity and completeness.

Example: "Patients who underwent GRA (CPT-27122) were identified through electronic medical records. Inclusion criteria were primary GRA for infected hip secondary to injection drug use; patients with secondary GRA or preexisting THA were excluded."

Have revised according to recommendations above.

“Following Institutional Review Board approval (JHM IRB00391703), retrospective chart review was performed at 2 United States academic trauma centers between January 2015 to December 2023. This investigation was conducted in accordance with the Declara-tion of Helsinki of 1975. Patients who underwent GRA (CPT-27122) were identified through electronic medical records. Inclusion criteria were primary GRA for infected hip secondary to injection drug use. Patients with secondary GRA or preexisting THA were excluded.”

Surgical Technique Description:

The surgical technique section is verbose and could benefit from being more concise and focused.

Example: "All patients received a posterior hip approach, with a lateral window approach used if infection extended into the pelvis. The decision to use an antibiotic spacer was made by the attending surgeon based on bone integrity post-debridement."

Have created a new sub-heading of Surgical Technique and made the section more concise:

“All patients received a posterior approach to the hip, with a lateral window approach if the infection extended into the pelvis. The decision to use an antibiotic cement spacer was made by the attending surgeon based on bone integrity of the acetabulum and proximal femur after debridement. Figures 1 and 2 illustrate severe bony involvement that precluded spacer placement, whereas Figure 3 an antibiotic spacer. The use of local antibiotic therapy was also at the discretion of the surgeon at the time of GRA. All surgeons took intraoperative deep cultures with fresh instruments at the time of GRA. A minimum of three samples were obtained, allowing for aerobic, anaerobic, and fungal cultures.”

Figures and Examples:

The reference to figures within the text can be distracting and is better placed in figure legends or separate explanations.

Example: "Figures 1 and 2 illustrate cases where severe bony involvement precluded spacer placement, while Figure 3 shows a patient with an antibiotic hip spacer."

Objective Outcome Measures:

The description of objective outcome measures should be clear and organized, with a focus on key outcomes.

Example: "Primary outcomes included mortality, conversion to THA, revision THA, ambulation status, and infection resolution. Secondary outcomes included patient demographics, social history, and medical comorbidities."

Patient Demographics and Comorbidities:

The description of demographics and comorbidities should be precise and focused, avoiding repetition.

Example: "Demographics collected included age, BMI, and gender. Comorbidities included chronic HCV, history of endocarditis, and psychiatric disorders."

Have now created two sub-headings, Primary Outcomes and Secondary Outcomes, and revised the sections to be more concise.

Primary Outcomes

The primary outcomes included mortality, conversion to THA, revision THA, ability to ambulate at final follow-up, and complete resolution of infection. A revision THA was defined as any procedure requiring a return to the operating room for a complication of the THA, including polyethylene exchange. Complete resolution of infection was defined as the eradication of infection with no further radiographic, laboratory, or clinical evidence of continued infection in the absence of any suppressive antibiotics. Laboratory studies rou-tinely ordered for infection at both institutions included CRP, ESR, and CBC.

Secondary Outcomes

Secondary outcomes included patient demographics, social history, and medical comorbidities. Demographics collected included age, body mass index (BMI), and gender. Social history included active injection drug use at initial hospitalization (defined as in-jection drug use within one week of hospitalization), tobacco use, and stable housing (de-fined as having a permanent place of residence). Medical comorbidities included chronic hepatitis C virus (HCV) infection, history of endocarditis, paraplegia, other diagnosed psychiatric disorders (defined as any formally diagnosed psychiatric disorder in the med-ical chart other than SUD), and the presence of systemic inflammatory response syndrome (SIRS), as per the Sepsis-3 consensus definitions.[20]”

PROMs Collection:

The description of PROMs should be detailed but concise, explaining the collection process and tools used.

Example: "PROMs were collected via phone-administered surveys with verbal consent. Measures included the EQ-5D-5L, NRS for pain, and Oxford Hip Score, with paraplegic and deceased patients excluded from collection."

Statistical Analysis:

The statistical methods should be described clearly, with an emphasis on the tests used and the rationale for their selection.

Example: "Continuous variables were assessed for normality using Shapiro-Wilks tests. Comparisons were made using Mann-Whitney tests for non-normal distributions or T-tests for normal distributions. Categorical variables were compared using Fisher’s exact test, with an α value of 0.05."

Have now revised to as follows:

“Continuous variables were compared with the Mann-U Whitney tests for non-normal distribution, or T-tests for normal distribution. Continuous variables were reported as mean ± standard deviation (SD) or median [interquartile range (IQR)] as appropriate. Categorical variables were compared with Fisher’s exact test. An α value of .05 was used.”

For Conclusions:

Conciseness and Clarity:

The conclusions should be succinct and clear, summarizing the key findings and their implications without unnecessary detail.

Example: "GRA in patients with septic hip arthritis secondary to injection drug use is associated with high mortality (27%) and low THA conversion rates (44%)."

Emphasis on Key Findings:

Ensure the emphasis is placed on the most significant findings and their implications for clinical practice.

Example: "Among those who underwent THA, 75% required revisions due to infections related to continued drug use."

Clinical Recommendations:

Recommendations should be clear and based on the findings, avoiding overly strong or ambiguous statements.

Example: "Due to the high risk of infection, second-stage total hip replacement is not recommended for patients with ongoing drug use."

Impact of Antibiotic Spacer:

The impact of using an antibiotic spacer should be clearly stated, emphasizing its association with infection resolution.

Example: "The use of an antibiotic spacer at the time of GRA was associated with improved infection resolution rates after a single admission."

Call for Further Research:

The need for further research should be highlighted, specifying the type of studies required.

Example: "Further large-scale, prospective studies are necessary to determine the optimal management strategies for this patient population."

Grammatical Precision:

Ensure grammatical accuracy and professional tone.

Example: "Further larger-scale studies" can be rephrased to "Further large-scale studies" for better readability.

Have now revised the conclusion in accordance with all of the recommendations above.

“GRA in patients with septic hip arthritis secondary to injection drug was associated with high mortality (27%) and low THA conversion rates (44%). Among those with THA conversion, 75% required revisions primarily due to infections related to continued drug use. Due to the high risk of infection, THA conversion is not recommended for patients with ongoing drug use. The use of an antibiotic spacer at the time of GRA was associated with improved infection resolution after a single admission. Larger prospective studies are necessary to determine the optimal management of these patients.”

Thank you again for your comments!

Sincerely,

Authors

Reviewer 2 Report

Comments and Suggestions for Authors

1. Kindly adhere to the EQUATOR reporting guideline for retrospective studies.

2. Please provide a filled-in checklist of reporting retrospective studies pertaining to the items listed in the report.

3. Multiple times several statistical tests have been employed. Hence, the type 1 error risk is inflated. Please use methods like Bonferroni's correction and redo the entire statistical inference accordingly.

4. How was the sample size estimation done? Elaborate the assumptions and also estimate the post-hoc power with the accrued data.

5. Table 3 represents the mean and SD with values collected in 2 patients in each group and this should be definitely reconsidered. This reviewer advises the authors to consider representing the individual values and do not attempt in carrying out any statistical analysis for QoL variables.

Comments on the Quality of English Language

Minor editing is required.

Author Response

Dear Reviewer 2,

Thank you for reviewing our manuscript and the thoughtful comments. We believe these comments have greatly improved the clarity of our manuscript. Our responses to the comments are below. We have made major revisions to almost every paragraph, and thus have not highlighted all the changed text. However, all the changes are tracked in the manuscript. We hope that you find our responses satisfactory.

  1. Kindly adhere to the EQUATOR reporting guideline for retrospective studies.

Thank you for pointing this out, we have now included a completed STROBE Checklist for Retrospective Cohort Studies, which also shows where we have included all the items in the manuscript.

  1. Please provide a filled-in checklist of reporting retrospective studies pertaining to the items listed in the report.

Thank you for pointing this out, we have now included a completed STROBE Checklist for Retrospective Cohort Studies, which also shows where we have included all the items in the manuscript.

  1. Multiple times several statistical tests have been employed. Hence, the type 1 error risk is inflated. Please use methods like Bonferroni's correction and redo the entire statistical inference accordingly.

Thank you for pointing this out. While we understand the role of p-value adjustment for multiple correction (ex. Bonferroni correction), we do not believe that it should be performed for our study for the following reasons:

  • Given our extremely small sample size, at baseline, the Type 2 error far outweighs the Type 1 error in this study. Thus, despite multiple testing, we believe that there was a relatively low risk of Type 1 error because the power is so low for this study.
  • This is a retrospective cohort study, and thus is inherently exploratory. The significant findings are intended to spur larger studies, and thus we believe that accepting a slightly greater Type 1 error risk from not performing P-value correction is more valuable than adjusting for multiplicity and inflating the already severely elevated Type 2 error risk. This also contrasts with other large-scale exploratory studies, such as GWAS “fishing expedition” studies, that have far more comparisons and sample size, which makes correction for multiplicity warranted. [1-4]

Therefore, while performing the Bonferroni correction is not technically demanding, we do not believe it is appropriate for this small retrospective cohort study. We have added further justification of our choice not to use any p-value correction in the Statistical Analysis section of the methods.

“P-value multiplicity correction was not performed for the two reasons: 1) the small sample size of this study results in a far greater risk for Type 2 Error as opposed to Type 1 error, even with multiple testing. 2) This retrospective cohort study was inherently exploratory, and its findings are meant to spur larger prospective studies.”

References:

  1. Jafari M, Ansari-Pour N. Why, When and How to Adjust Your P Values? Cell J. 2019 Jan;20(4):604-607. doi: 10.22074/cellj.2019.5992. Epub 2018 Aug 1. PMID: 30124010; PMCID: PMC6099145.
  2. Rothman KJ. No adjustments are needed for multiple comparisons. 1990 Jan;1(1):43-6. PMID: 2081237.
  3. Streiner DL, Norman GR. Correction for multiple testing: is there a resolution? 2011 Jul;140(1):16-18. doi: 10.1378/chest.11-0523. PMID: 21729890.
  4. Streiner DL. Best (but oft-forgotten) practices: the multiple problems of multiplicity-whether and how to correct for many statistical tests. Am J Clin Nutr. 2015 Oct;102(4):721-8. doi: 10.3945/ajcn.115.113548. Epub 2015 Aug 5. PMID: 26245806.
  5. How was the sample size estimation done? Elaborate the assumptions and also estimate the post-hoc power with the accrued data.

There was no sample size estimation done. We captured as many patients as we could, based on when the lead trauma attending started seeing patients at the institution. This was the furthest we could go back in our electronic medical record search using those CPT codes with those specified surgeons (the authors of the study).

We disagree with performing any post-hoc power analysis, especially for any retrospective studies, as this analysis is conceptually flawed and is misleading. [1-4] In our limitations, we have highlighted the low sample size of our study, which should caution readers about the findings of this study.

“The small sample size of this study is a severe limitation, impacting the generalizability and statistical power of the study’s findings.”

References:

  1. Zumbo, B. D., & Hubley, A. M. (1998). A Note on Misconceptions Concerning Prospective and Retrospective Power. Journal of the Royal Statistical Society. Series D (The Statistician), 47(2), 385–388. http://www.jstor.org/stable/2988675
  2. Heinsberg LW, Weeks DE. Post hoc power is not informative. Genet Epidemiol. 2022 Oct;46(7):390-394. doi: 10.1002/gepi.22464. Epub 2022 Jun 1. PMID: 35642557; PMCID: PMC9452450
  3. Zhang Y, Hedo R, Rivera A, Rull R, Richardson S, Tu XM. Post hoc power analysis: is it an informative and meaningful analysis? Gen Psychiatr. 2019 Aug 8;32(4):e100069. doi: 10.1136/gpsych-2019-100069. PMID: 31552383; PMCID: PMC6738696.
  4. Heckman MG, Davis JM 3rd, Crowson CS. Post Hoc Power Calculations: An Inappropriate Method for Interpreting the Findings of a Research Study. J Rheumatol. 2022 Aug;49(8):867-870. doi: 10.3899/jrheum.211115. Epub 2022 Feb 1. PMID: 35105710.

  1. Table 3 represents the mean and SD with values collected in 2 patients in each group and this should be definitely reconsidered. This reviewer advises the authors to consider representing the individual values and do not attempt in carrying out any statistical analysis for QoL variables.

Thank you for this, we have now revised this table to report the PROMs of the individual patients that responded. We have also now stated that no meaningful statistical analysis could be performed with such few patients.

“Given the low response rates, no statistical analysis was performed.”

Overall

N=4

Patient 3

(no Spacer)

Patient 4

(no Spacer)

Patient 13

(Spacer)

Patient 14

(Spacer)

EQ-5D Mobility

3.3 ± 1.3

5

2

3

3

EQ-5D Self-care

1.3 ± 0.5

2

1

1

1

EQ-5D Activity

2.3 ± 1.5

4

1

3

1

EQ-5D Pain

3.0 ± 1.4

5

2

3

2

EQ-5D Anxiety

1.8 ± 1.0

3

1

1

2

NRS Least pain

3.0 ± 2.5

5

2

5

0

NRS Most pain

6.8 ± 2.5

8

8

8

3

NRS Average pain

3.8 ± 3.5

8

2

5

0

Oxford Hip score

30 ± 13

16

34

24

46

Thank you again for your comments!

Sincerely,

Authors

Reviewer 3 Report

Comments and Suggestions for Authors

some important point to modify, all reference citation should be before comma at the end of the phrases.

Introduction:

this phrase is not necessary (or Girdlestone procedure/situation)

line 38 : add reference

the article should be devised in introduction - materials and methods - results - discussion - conclusion    :   the materials and methods were placed after discussion

this should be added , with a clear inclusion exclusion criteria + flowchart (what happened to the 51 patients ? why they were excluded)

surgical intervention and technique should be added , as it was not mentioned , was reaming of acetabulum done ? as cartilages should be removed during girdlestone to decrease recurrence

you don't need each time to write The mean ± SD then you write the number ,   this should be written directly (for example : the mean age was xx ±xxx)  and this should be modified overall the manuscript and in the abstract

what is the minimum maximum follow up ? as standard deviation is was high ?  what was the minimum follow-up ? as most infections studies should have a minimum of 2 years

what was the criteria of choosing spacer vs no spacer

what type of spacer ? what antibiotic and dose was used in the spacer?

provide a picture of spacer used  (as images seems that a metallic prosthesis was used and covered with cement instead of full spacer with cement .

why authors did not use functional outcome such as merle and aubigne score ?

what was the rule of MRI in the opinion of authors as it was mentioned in the results section and never appeared again

i believe that p value measure have no value as few number of patients 10 vs 5 

lastly : why 10 authors ?  only authors who had a vital contribution should be associated

Author Response

Dear Reviewer 3,

Thank you for reviewing our manuscript and the thoughtful comments. We believe these comments have greatly improved the clarity of our manuscript. Our responses to the comments are below. We have made major revisions to almost every paragraph, and thus have not highlighted all the changed text. However, all the changes are tracked in the manuscript. We hope that you find our responses satisfactory.

some important point to modify, all reference citation should be before comma at the end of the phrases.

Thank you for pointing this out, we have now ensured the references are placed before commas where appropriate. Other references are still placed after the period at the end of a sentence per journal requirements.

Introduction:

this phrase is not necessary (or Girdlestone procedure/situation)
Have now removed this phrase. The introduction has also been majorly edited to make it more concise.

line 38 : add reference

This sentence has now been removed, along with major edits throughout the manuscript to make the paper more concise.

the article should be devised in introduction - materials and methods - results - discussion - conclusion    :   the materials and methods were placed after discussion

Although we prefer our articles to be formatted like this, the current formatting is per the Journal’s requirements. The Journal requires introduction, results, discussion, methods, and conclusion.

this should be added , with a clear inclusion exclusion criteria + flowchart (what happened to the 51 patients ? why they were excluded)

We have now included a detailed diagram that shows the excluded patients. (Figure 1)

surgical intervention and technique should be added , as it was not mentioned , was reaming of acetabulum done ? as cartilages should be removed during girdlestone to decrease recurrence

Reaming was only done if the articulating hip spacer was placed. Otherwise, the cartilage of the acetabular joint was completely removed with curettes. This has now been added to the methods.

“Acetabular reaming was only performed if a spacer was placed, otherwise cartilage was fully removed via curettage.”

you don't need each time to write The mean ± SD then you write the number ,   this should be written directly (for example : the mean age was xx ±xxx)  and this should be modified overall the manuscript and in the abstract
This has now been revised throughout the manuscript.

what is the minimum maximum follow up ? as standard deviation is was high ?  what was the minimum follow-up ? as most infections studies should have a minimum of 2 years

The minimum follow-up was 1 month (patient died) and maximum follow-up was 69 months (~6 years). When excluding the patients that died, the minimum follow-up was 6 months. For outcomes analysis, we used a minimum follow-up of 1 month, which is the standard cut-off for most orthopedic trauma studies.

We have now added the ranges to the results.

“The mean follow-up was 25±20 months (range 1-69 months) (32±20 months, range 6-69, when excluding deceased patients).”

what was the criteria of choosing spacer vs no spacer

The decision to use a spacer was dependent on the subjective assessment of enough bony stock in the acetabulum and proximal femur after debridement. This is now clarified in the methods:

“The decision to use an antibiotic cement spacer was made by the attending surgeon based on bone integrity of the acetabulum and proximal femur after debridement.”

what type of spacer ? what antibiotic and dose was used in the spacer?

Either a Depuy Synthes Prostalac Hip System or a Zimmer Biomet Taperloc or Echo stem with antibiotic cement was used. 40g Palacos cement with 3g of vancomycin and 1.2g of tobramycin was used in all cases for cement fixation of the articulating spacer. We have now added this to the surgical technique section in the methods.

“Either a Depuy Synthes (Raynham, Massachusetts) Prostalac Hip System or a Zimmer Biomet (Warsaw, Indiana) Taperloc or Echo stem with antibiotic cement was used. 40g Palacos (Heraeus Medical, Hanaus, Germany) cement with 3g of vancomycin and 1.2g of tobramycin was used in all cases for cement fixation.”

provide a picture of spacer used  (as images seems that a metallic prosthesis was used and covered with cement instead of full spacer with cement .

Antibiotic cement articulating spacers were used in patients with spacers, not fully cement spacers. Figure 3 shows the articulating antibiotic cement spacer (Deput Synthes Prostaloc) that was used in all patients that had received a spacer. Have now clarified this in the Figure 3 caption.

why authors did not use functional outcome such as merle and aubigne score ?

We chose PROMs that were commonly used in THA literature [1], as well as ones that we were familiar with the interpretation of the scores. Additionally, given how difficult it was to get patients return to clinic, we chose scores that could be collected by phone. The Merle d'Aubigné and Postel Method and Modified Merle d'Aubigné and Postel Method require range of motion, which was not feasible to obtain.

References:

  1. Vajapey SP, Morris J, Li D, Greco NG, Li M, Spitzer AI. Outcome Reporting Patterns in Total Hip Arthroplasty: A Systematic Review of Randomized Clinical Trials. JBJS Rev. 2020 Apr;8(4):e0197. doi: 10.2106/JBJS.RVW.19.00197. PMID: 32539265.

what was the rule of MRI in the opinion of authors as it was mentioned in the results section and never appeared again

MRI was only obtained if the extent of infection was unclear on CT imaging to help better guide preoperative planning. 3 patients got MRIs preoperatively. In those cases, it also confirmed the infection. We have clarified this in the Results now, as CT imaging with contrast is our primary imaging modality (other than plain radiographs) for infection. Every patient got CT imaging preoperatively.

“Inclusion was based on the documentation of injection drug use in patient’s history and radiographic, CT imaging, and/or MRI signs of infection”

i believe that p value measure have no value as few number of patients 10 vs 5 

Thank you for this point, we also agree that our small sample size limits the validity and generalizability of our analysis. We have highlighted this in our limitations. “The small sample size of this study is a severe limitation, impacting the generalizability and statistical power of the study’s findings.”

lastly : why 10 authors ?  only authors who had a vital contribution should be associated

Many of the authors (AH, RS, GO, BS) were the attending surgeons that performed these cases at the institutions, and thus provided the patients for this study. JF and JC were the expert consultants that provided guidance for management of the patients, but also provided expert review in writing this manuscript. DG, OC, and ZE were the research fellows/medical students that performed data collection and called patients over several months in attempt to obtain the PROMs. All authors contributed to writing the draft, review, and also approval before submission. Detailed contributions are also attached to the manuscript.

Thank you again for your comments!

Sincerely,

Authors

Round 2

Reviewer 1 Report

Comments and Suggestions for Authors

Dear Authors,

I have carefully reviewed the revisions you made to your manuscript following my feedback. I am pleased to note that the changes have substantially improved the overall quality of the paper.

The adjustments to the title and abstract have made the focus of your study much clearer and more concise. The enhanced clarity in the methodology, particularly the detailed description of patient selection and data analysis, strengthens the robustness of your findings. Additionally, the restructuring of the results section has provided a more logical and accessible presentation of your data, which is crucial for the readers' understanding.

The discussion section is now more focused, with a clearer interpretation of your findings in the context of existing literature. Your emphasis on the implications for clinical practice, especially regarding the use of antibiotic spacers and the challenges associated with treating injection drug users, is well articulated and valuable.

In light of these improvements, I believe that the manuscript is now well-prepared for publication. Your work has addressed the initial concerns effectively, and the resulting manuscript offers significant contributions to the field.

Thank you for your diligent work on these revisions.

Sincerely,

Reviewer 2 Report

Comments and Suggestions for Authors

Thank you for the revision.

Reviewer 3 Report

Comments and Suggestions for Authors

Authors have adressed all issues and modified the study with major modifications which was appropriate and satisfying. 

i have no other issue to add